# **HoloTrack: In-Situ Holographic Particle Tracking Velocimetry of Cloud Droplets**

Birte Thiede<sup>1,2</sup>, Freja Nordsiek<sup>1</sup>, Yewon Kim<sup>1</sup>, Eberhard Bodenschatz<sup>1,2,3</sup>, and Gholamhossein Bagheri<sup>1</sup>

**Correspondence:** Gholamhossein Bagheri (gholamhossein.bagheri@ds.mpg.de)

**Abstract.** We present HoloTrack, a novel, fully autonomous measurement system designed to capture three-dimensional (3D) cloud droplet data and provide detailed insights into droplet dynamics, their spatial distribution and velocity. The HoloTrack system integrates a high-accuracy holographic imaging system with environmental sensors, including pitot tubes for airflow measurements, and a navigation system. Designed for deployment on airborne platforms like the CloudKite and hence having a compact and autonomous design, HoloTrack is also ideally suited for deployment in laboratory or ground-based environmental research. The system records up to 25 hologram pairs per second (50 holograms per seconds with the resolution of 65 megapixels), each of which provides two independent measurements of droplet position, size, and shape and measures individual droplet velocities. With laboratory tests we confirmed, that the holographic system reliably detects particles down to 10 µm, within a sample volume of 17 cm<sup>3</sup> (1.84 cm × 1.84 cm × 5 cm) of each hologram. For a recorded hologram pair with mean displacement of 0.5 cm caused by e.g. an inter-frame time of 500 µs and a mean velocity of 10 m/s, this results in 21.5 cm<sup>3</sup> combined volume, where particle position and size is sampled and 12.3 cm<sup>3</sup> overlapping volume where the two-frame particle velocimetry can be applied to resolve individual droplet velocities. Reliable sub-volumes for measuring droplets at different yaw angles, to account for the influence of the instrument body are further defined. The droplet velocity in longitudinal and vertical direction is measured with errors of less than 0.07 m/s for inter-frame times of 500 µs. The transverse velocity is less accurate with errors in the range of 0.1-0.5 m/s, depending on the position of the particles in the sample volume. Nevertheless, the flexible timing allows the adjustment to different displacements to optimize the overlapping volume and 3D velocity uncertainties according to the needs of the experiment. A series of laboratory tests and a maiden flight tests validated the system's capabilities, characterizing the detection, robustness, automation and its ability to measure droplet dynamics. HoloTrack's unique combination of holographic particle measurements including capturing their velocities makes it a powerful tool for advancing our understanding of cloud microphysics, including droplet spatial distribution, collision-coalescence, entrainment, and turbulent mixing processes.

<sup>&</sup>lt;sup>1</sup>Max Planck Institute for Dynamics and Self-Organization (MPI-DS), Am Faßberg 17, 37077 Göttingen, Germany

<sup>&</sup>lt;sup>2</sup>Faculty of Physics, University of Göttingen, Friedrich-Hund-Platz 1, 37077 Göttingen, Germany

<sup>&</sup>lt;sup>3</sup>Laboratory of Atomic and Solid State Physics, Cornell University, 523 Clark Hall, Ithaca, NY 14853, USA

## 1 Introduction

Clouds have a significant influence on weather and climate and play a crucial role in the Earth's radiative energy budget. Cloud properties emerge from a complex interplay between microphysical processes, such as droplet size distribution, and dynamics and large-scale cloud dynamics. Microphysics both respond to and influence the thermodynamic environment and turbulent motions within clouds (Shaw, 2003) with droplet size evolution closely linked to turbulent flow and the history of entrainment and mixing (Grabowski and Wang, 2013). Understanding these processes remains a challenge due to the multi-scale nature of clouds, from droplet-level physics to large-scale atmospheric dynamics (Bodenschatz et al., 2010)which is why clouds remain the most uncertain climate feedback (Forster et al., 2021). Resolving individual cloud droplets is not possible via remote sensing (Grosvenor et al., 2018), so optical probes are commonly deployed. Generally these can be divided into two groups (see discussions about probes in e.g. Beals, 2013; Korolev et al., 2017): traditional probes measuring a single particle at a time, probing a quasi-1D volume and camera based measurements that sample droplets within large localized two-dimensional ((e.g. Particle Image Velocimetry, PIV, in MPCK<sup>+</sup> Schlenczek et al., 2025) and (Bertens et al., 2021)) or with holography even three-dimensional cloud volumes with each sample.

Holographic instruments have successfully measured cloud droplets in-situ for over 30 years (Brown, 1989), including current instruments like HOLODEC (Fugal and Shaw, 2009; Spuler and Fugal, 2011), HALOHolo (Schlenczek, 2018; O'Shea et al., 2016; Lloyd et al., 2020), HOLIMO (Henneberger et al., 2013; Ramelli et al., 2020) and the Advanced Max Planck CloudKite Instrument (MPCK<sup>+</sup>) (Schlenczek et al., 2025; Thiede et al., 2025a). These systems allow comprehensive and more localized statistical analysis of cloud microphysical properties, such as concentration, local size distribution (Fugal and Shaw, 2009; Allwayin et al., 2024), and spatial characteristics like droplet clustering in full three dimensions (Borrmann et al., 1993; Larsen et al., 2018; Glienke et al., 2020; Thiede et al., 2025a) or analyze the cloud mixing behavior (Beals et al., 2015; Desai et al., 2021). Recent studies confirm the intermittent, "patchy" nature of clouds (Jameson and Kostinski, 2001; Allwayin et al., 2024; Thiede et al., 2025a), with significant variations over small horizontal distances, highlighting the importance of large-volume, localized imaging provided by holography. Current holographic instruments measure 3D droplet positions and cross-sectional sizes for particles typically larger than 6–10 μm within sample volumes of the order of 10cm<sup>3</sup>. Despite the described advantage of these measurements and recent achievements of holographic cloud droplet measurements, a key aspect of cloud microphysics remains largely inaccessible: droplet dynamics.

Holographic particle velocimetry has been successfully used in laboratory fluid dynamics contexts (Meng and Hussain, 1991; Hinsch, 2002; Tao et al., 2002; Hinsch and Herrmann, 2004; Meng et al., 2004; Svizher and Cohen, 2006, just to name a few), the high true-air-speed in airborne measurements and the constraints in camera pixel size and field of view to resolve the small cloud droplets, makes it a challenge for in-situ cloud measurements. Even the MPCK<sup>+</sup>instrument, with the highest airborne holographic sampling rate (75 Hz) on a tethered aerostat, captures entirely different sample volumes in consecutive holograms, preventing assessment of droplet dynamics. MPCK<sup>+</sup>incorporates a 2D PIV system to measure droplet motion in a quasi-2D laser sheet (4 mm thick), but it cannot capture individual droplet velocities and their size.

5 We have developed HoloTrack, the first holographic cloud droplet tracking velocimetry instrument that simultaneously mea-

sures droplet size and 3D velocity in a large, localized sample volume. By combining the low true-air-speed of the Max Planck CloudKite platform with advanced camera technology and precise timing, HoloTrack captures hologram pairs with overlapping volumes, enabling droplet tracking in three dimensions. Figure 1 illustrates the principle: the system records a hologram pair ( $H_1$  and  $H_2$ ), and, based on the inter-frame time  $\Delta t$  and mean true-air velocity u, droplets captured in the upstream part of  $H_1$  are also captured downstream in  $H_2$  (blue droplets in Figure 1). Matching droplets across the hologram pair allows computation of 3D velocities, though the z-component is associated with higher uncertainties. As holography also captures the cross-sectional size of particles, HoloTrack is not just limited to measuring droplets but also ideally suited for capturing non-spherical particles. In this paper, we present the design considerations and technical details involved in building HoloTrack. Through a test flight, wind tunnel droplet measurements, and our static test target (i.e. the CloudTarget introduced in Thiede et al., 2025b) we comprehensively characterize detection, sizing, position and velocity uncertainties, outline potential improvements, and highlight HoloTrack's capabilities.

## 2 Instrument Design

# 2.1 Mechanical Design

The HoloTrack planned design and the instrument that was finally manufactured are shown in Fig.2. With dimensions of 130 cm × 38 cm × 20 cm (excluding removable legs, battery holder and stabilizer fin), the HoloTrack instrument box maintains a moderate size, making it suitable for various laboratory setups and transportable within the Mobile Cloud Observatory for deployment on the CloudKite. The instrument consists of the main body that houses all key measurement logging and automation instruments, including two computers (see Section 2.4) and the two upstream-oriented "arms" of the holographic system. This general design is inspired by previous holographic systems used for cloud droplet measurements such as Halo-Holo and HoloDEC (Spuler and Fugal, 2011; Schlenczek, 2018). Termed the "Laser Arm", one arm encapsulates the optics for laser beam alignment, expansion, and collimation. The second arm "Camera Arm" accommodates the camera that records the holograms without any lens.

HoloTrack was designed to have a stable laser beam-path system to avoid the need for realignment of the optics post-transportation or experiments. Therefore, both the laser and all optical components are mounted onto the single solid 2 cm thick base-plate with several screws to avoid any movement including vibrations. Aluminum was chosen as the material for the main instrument structure, which was optimized for weight by incorporating cutouts or a width reduction in honeycomb pattern in most structural components. The instrument can be easily handled and carried by two persons. The instrument box features side windows for the visual inspection of electronic connectors and status LEDs to enable error identification. A top window with integrated touchscreen allows operators to use the custom-made graphical measurement control software (written in Python Tkinter) and observe measurement status. Designed to withstand flight in precipitating clouds, the instrument box is constructed to be fully sealed and waterproof. The front of the instrument box as well as the arms of the holographic systems are designed to minimize the aerodynamic disturbance to the flow around them and, therefore, low aerodynamic disturbance in the sampling volume. This also ensures better alignment with the mean wind when attached to the CloudKite tethered balloon

and minimal influence of the instrument body on the sample volume. The arm and front covers are 3D-printed and shown in blue in Figure 2. The hologram arms, long relative to the cross section of the instrument box, position the holographic sample volume at a large distance from the instrument body to minimize the impact of the bluff-body effect. The design of the tips of the holographic arms is inspired by the tips discussed in (Korolev et al., 2013) to avoid particle shattering. The holographic system's optical axis, which in our convention is the z-direction, is orientated horizontally, leading to a vertical orientation of the windows on the camera and laser arms, chosen to impede dust and water accumulation. For in-flight use HoloTrack is further equipped with a holder for the battery in the back and a stabilizer fin for mean-flow orientation as shown in the photo in Fig. 2. Acting as a heat sink, the base plate along with the honeycomb pattern effectively disperse heat into the surrounding flow. Nevertheless, the HoloTrack is equipped with two Peltier Elements for automatic temperature control for operations under more extreme temperatures.

# 2.2 Holographic Setup

In the design of the HoloTrack holographic instrument setup, we needed to consider various factors for accurate measurements of cloud droplets. Specifically, the smallest detectable droplets are desired to be around 6  $\mu$ m, and typical expected velocities are on the order of 10 m/s. Particularly the detection of smaller droplets at higher depths within the holographic volume is limited by the cameras pixel size  $d_{pixel}$ , the field of view  $N_x d_{pixel} \times N_y d_{pixel}$  in combination with the illumination wavelength  $\lambda$ . Therefore, the combination of illumination source and camera needs to be carefully chosen. Particles with a diameter smaller than two pixels can generally not be resolved using our standard hologram processing techniques with wavefront reconstruction via the Huygen-Fresnel kernel (Fugal et al., 2009). In addition to a small pixel size, the camera sensor should also have a large cross-sectional field of view. This feature is needed to resolve small droplets at larger depths, as the crucial particle information carried by diffraction patterns in holograms spreads over a large x-y extent for small particles located farther from the camera sensor (detailed description in Fugal et al. (2009); Thiede et al. (2025b)).

HoloTracks holographic system, specifically, also demanded a camera with a high frame rate and flexible exposure timing options. A high frame rate is generally desired in in-situ holography to record the localized holographic samples at high spatial frequency. The XIMEA CB654MG-GP-X8G3 camera, with small pixel size of 3.2  $\mu$ m and large field of view of  $22.4 \times 29.9$  mm, has flexible timing and therefore allows for a short inter-frame time of sub-milliseconds within hologram pairs to allow particle tracking. The small pixel size also means that no lens is required for the camera, which simplifies the design and significantly reduces the weight. The exterior-facing side of the camera window is at reconstructed z=2.5 cm and the exterior-facing side of the laser window at z=22 cm. The camera is operated at 8-bit.

For illumination a suitable coherent light source is needed. The laser pulse energy should be high enough to reach approximately 50% of the full well capacity FWC in the camera after expansion and transmission through all optical components (see Section 2.2.1) for optimal signal-to-noise ratio. The desired energy density can therefore be expressed as

$$120 \quad e_d = \frac{0.5FWC}{ge(\lambda)} \frac{hc}{\lambda} d_{px}^2 \,, \tag{1}$$

where  $qe(\lambda)$  is the quantum efficiency of the camera at the laser wavelength  $\lambda$ , h is Plancks constant and c the speed of light. The required pulse energy for the laser then depends on this desired energy density, the expansion of the beam up to a diameter of  $d_{laser}$  at the camera sensor and the combined transmission of all optical components between laser head and camera  $T_{all}$ :

$$E_{pulse} = e_d T_{all} \pi \left( \frac{d_{laser}}{2} \right) . \tag{2}$$

To achieve an even illumination across the whole camera senors, the laser was expanded to  $d_{laser} \approx 2 \times d_{camera}$ , where 125  $d_{camera}$  is the sensors diagonal (this is further discussed in Section 2.2.1). We chose a green laser with 532 nm wavelength, the Explorer One XP (Newport Spectra-Physics). The laser offers flexible timing options including burst operation, a compact size and adequate pulse energy. While depth resolution decreases with wavelength, the chosen XIMEA CB654MG-GP-X8G3 camera has high quantum efficiency for 532 nm, hence the green laser being a good fit. The achieved z- and particle diameter-130 dependent detection is tested in Section 3.2. Lastly, the separation of the window in the laser arm and the camera arm window determines the effective sample volume dimension in z. Though ideally, a larger sample volume is always preferred, we settled for a separation of 19.5 cm. This is because the size of the smallest resolvable droplet decreases with an increase in z. Additionally, the shadow density increases with larger z-component of the sample volume. According to empirical results by Royer (1974) hologram quality deteriorates for SD > 1% and a theoretical upper bound limit for which holography becomes unsuitable is given by Meng et al. (1993) with approximately 40% (G=1 in Meng et al., 1993). With a 19.5 cm z-extent, the 135 shadow density is  $SD = \pi 19.5 \text{cm}/4 \sum_i n_i d_i^2$  which would be 1%, 3%, 7% for  $500 \text{ cm}^{-3}$  monodisperse droplets of  $10,20,30 \text{ }\mu\text{m}$ respectively. Hologram quality is therefore expected to only be strongly affected in conditions with exceptionally large number concentration and droplet sizes.

While the camera is able to reach frame rates up to 71 fps, we typically operate it at 50 fps i.e. 25 hologram pairs. At a nominal mean velocity of 10 m/s this yields a three-dimensional sample every 40 cm horizontal distance resolving the cloud at sub-m resolution.

# 2.2.1 Laser Optics

We aimed to design a holographic system with collimated light to establish a rectangular sample volume. For this, on the laser side, the laser beam has to be expanded up to at least the sensor diameter  $d_{laser} > d_{sensor} = 3.7$  cm to illuminate the full sample volume. To optimize for near-constant detection efficiency in the cross-section (x-y) even illumination of the sensor is ideal. A straightforward solution is to expand the beam beyond the necessary diameter and utilize only the center of the Gaussian beam. In HoloTrack this expansion has to be achieved over a beam path of approximately 45 cm based on the current design setup. We accomplished the beam's expansion and collimation using a set of four aspheric lenses with focal lengths  $f_1 = 8$  mm,  $f_2 = 10$  mm,  $f_3 = 32$  mm and  $f_4 = 100$  mm as shown in Figure 3. The laser beam is emitted from the inside of the laser head with a small divergence angle. First, with an adjustable alignment mirror the beam is aligned into the center of the laser arm. The first three lenses amplify the divergence angle of the beam. The beam is spatially filtered with a 15  $\mu$ m-pinhole, which is approximately 1.5 times the size of the beam waist, positioned in the first focus behind the  $f_1 = 8$  mm aspheric lens. Towards the end of the laser arm, we placed the final fourth aspheric lens that collimates the beam when it has expanded to

a theoretical diameter of approximately 8 cm. However, in practice, the aperture trims the beam to a final size of about 5 cm. After collimation, a mirror guides the beam into the sample volume.

Collimation was tested with different methods in the process of optimizing it and in CloudTarget evaluation we saw a negligible bias in the random position error of z (see Section 3.2). The beam intensity is adjusted with an absorbent neutral density filter to optimize the mean intensity in the holograms to about 50% of the well-depth. Given the timing constraints when operating the camera with minimal inter-frame times, the second frame of each hologram pair has a long exposure (see Section 2.2.2 for details). Consequently, the collection of ambient sunlight by the camera needs to be limited. We accomplished this by using a bandpass filter with a 10 nm bandwidth centered at 532 nm and a liquid crystal shutter (FOS-AR, LC-TEC) in front of the camera sensor. The shutter, operable by a voltage signal, can be set to be open (with a transmission of 80% for polarized light, opening time 35 ms) or closed (0.02% transmission) within 150  $\mu$ s (at 20°C, 350  $\mu$ s at 0°C) and can be operated down to temperatures of  $-10^{\circ}$ .

## **2.2.2** Timing

In the holographic system the timing of laser pulses, camera exposure and liquid crystal shutter is essential to successfully achieve short inter-frame times without measuring a high background intensity from the ambient sunlight. All the timings are controlled by a sequence generator developed by the in-house electronics department of the Max-Planck-Institute for Dynamics and Self-Organization (MPI-DS). The sequence generator has 8 output channels, where the voltage (4 outputs with 5 V, 4 outputs with 24 V) can be controlled in µs-steps. With the outputs the laser pulse bursts are triggered, the camera exposure times are defined and triggered and the liquid crystal shutter is set into an open or closed state.

What we call inter-frame time is not exactly the time between the frames i.e. the time between camera exposures but the time between the two laser pulses recorded in holograms  $H_1$  and  $H_1$  of a pair. The laser is running at a frequency  $f_l$  and is emitting  $n_P$  pulses per burst. The general idea to achieve accurate and short inter-frame times for tracking is that the first hologram  $H_1$  of each pair records the first laser pulse of each burst and the second frame records the  $n_p$ th pulse. The effective inter-frame time then is  $\frac{1}{f_l}(n_p-1)$ . A lower limit for effective inter-frame time is the minimal time between the end of one frame and the start of the second frame, the frame overhead time, which is stated to be  $28\mu s$  by the manufacturer Ximea. While minimal exposure time is technically 0.1 ms, the second exposure time needs to be equal or longer than the readout time of the first frame, which is related to the maximal frame rate  $t_{rd} \approx \frac{1}{f_{max,cam}}$ , where  $f_{max,cam} = 71 \text{Hz}$ . Hence, the first exposure  $H_1$  is set to be  $t_{H1} = 0.1$  ms but the second exposure  $H_2$  has to be  $t_{H2} \approx t_{rd} - \Delta t$ , where  $\Delta t$  is the time between the two exposures. At a wavelength of 532 nm the ambient sunlight collected with the camera, even with the 10 nm bandpass filter installed, would increase the background intensity to a level above the actual signal from the laser. Therefore the liquid crystal shutter is timed to close after 0.1 ms of the second exposure.

The holographic system timing in HoloTrack was optimized for a mean flow speed of 10 m/s for all measurements shown in this paper, but it can be easily modified for 1 m/s to 100 m/s. For the timing protocol used here with inter-frame time of 500 µs, the laser frequency is set to 80 kHz and is configured in burst mode emitting bursts of each 41 pulses 25 times per second. The first exposure stops about 6µs after the first pulse and the second exposure starts 6µs before the 41st pulse, which ensures only

a single laser pulse is recorded in each hologram. Therefore the effective inter-frame time is 500 $\mu$ s. The liquid crystal shutter is open for the whole duration of the laser pulse burst and closes  $\approx$ 0.1 ms after last laser pulse. According to the manual of the laser the inaccuracy for the laser frequency and therefore for our effective inter-frame time is less that 0.1% at the 80 kHz used in the described timing protocol to achieve 500 $\mu$ s. The triggering signals emitted by the sequence generator have to take the laser, shutter and camera delays into account and the LC-shutter requires a specific signal pattern to be in the open or closed state. For different timing schemes, it is practical to adjust the number of laser pulses in the burst and keep the frequency constant at 80 kHz, since first-pulse suppression settings and the neutral density filters in the beam path would otherwise need to be changed to compensate for the varying pulse energies at different repetition rates. Therefore, the timing can be freely adjusted by 12.5  $\mu$ s steps.

## 2.3 Measurement Instruments and Sensors

The HoloTrack instrument consists of several measurement systems, the main one certainly being the holographic particle tracking velocimetry system described above. Besides that, HoloTrack is equipped with two pitot Tubes for flow measurement. This includes a 1D pitot tube running at 100 Hz, where pressure is recorded and directly converted into velocities on the ADC (Air Data Computer by Simtec AG) and a 5-hole-pitot tube, running at 50 Hz, connected to the VectoDAQ which translates the pressured recorded in 5 angles into the three velocity components and flow angle of attacks.

The SBG Ellipse-N is an Inertial Navigation Unit (INU) providing information on orientation (roll, pitch, yaw), velocity, and position of HoloTrack through a combination of GPS and inertial data. This not only provides essential information about measurement location but also allows corrections of the measured velocities from the pitot tubes and the particle tracking velocimetry system for instrument motion. For redundancy a multi-band, centimeter-level GNSS receiver board (the simpleRTK2B by ArduSimple, integrating the u-blox ZED-F9P modules), is also installed on HoloTrack, along with three GPS antennas. It is however currently not operational due to USB-interface issues in the current version.

The OPC-N3 particle sensor can measure aerosols and small cloud droplets as a reference or potential trigger for the holographic system. HoloTrack is also designed to be equipped with the CDP2, which would provide reliable particle concentration and size distribution reference in a qausi-1D measurement. During the test flight and evaluation experiments shown below, no CDP2 was installed yet. In the cap of HoloTrack additional small-scale sensors (SHT40, BMP390, TMP117, BME688) are installed to measure quantities like temperature, pressure and relative humidity. See Table 1 for more details about these sensors.

#### 215 **2.4** Integration and Automation

HoloTrack is fully automated and can operate in two modes. In manual mode, an operator can start and stop holographic measurements using the graphical user interface on the mounted touchscreen. Alternatively, in trigger mode, measurements are initiated automatically based on altitude or particle concentration using devices such as the OPC-N3. By avoiding reliance on radio communications, which have caused problems in our previous instrument designs, the setup remains entirely autonomous.

The acquisition and automation system consists of two computers: the main computer controls the measurement status and logs data from all instruments listed in Table 1 except for holographic images. The camera of the holographic system is connected the "holo-computer", which logs only the holographic data. HoloTrack can be powered with a power supply in laboratory settings or with a battery (see Figure 2 bottom) for in-flight measurements. The IP67 25.6 V, 50 Ah LiFePO<sub>4</sub> battery, which includes its own battery management system, provides sufficient capacity for several hours of flight. With four 1 TB hard disks a full hologram capture run can store approximately 60,000 holograms in about 20 minutes of continuous operation. As soon as HoloTrack is powered on the main computer boots and the measurement program with the graphical user interface is opened. With this, all measurement systems (described in Section 2.3) except for holography are started and the recorded data is automatically logged on the main computer. We do currently see issues with connectivities of the sensors, likely cause by ground-loops, which leads to some intermittency in the data logging, leading to second-long gaps in the recorded data. Connection to sensors are checked continuously and once a missing sensor is back online, data acquisition continuous seamlessly. Due to laser safety considerations (see Section 2.5), as well as the system's high energy demands and substantial data production, the holographic system does not start automatically. Instead, it must be activated either manually through the graphical user interface or automatically triggered when operating in flight mode. This triggering is currently implemented to be caused by a certain barometric altitude. Before a flight on the CloudKite the cloud altitude can be determined by operators and set as a trigger limit. Since the OPC-N3 also measures particle count a triggering by this could also be implemented. The holographic system is turned on in 3 levels *Ready*, Arm and Acquisition. These levels can be selected manually or by a trigger and exist to prevent waiting times for start of acquisition due to minutes-long boot times of the holographic computer or temperature stabilizing time of the laser head. In the *Ready* state the camera and the holo-computer for hologram acquisition are turned on. The holographic capturing code starts up automatically on the holo-computer and as soon as the main computer can communicate with the holo-computer, the holographic state is *Ready*. The hologram acquisition code on the holographic computer would now save any incoming frames. For Arm the laser is turned on and is trying to reach a stable temperature. To reach the final Acquisition-state, where holograms are actually recorded, all interlocks are closed and the sequence generator is powered to send triggering signals to the laser, the camera and the liquid crystal shutter to follow the timing protocol described in Section 2.2.2. If only a brief interruption from hologram acquisition is planned a switch from Acquisition to Arm and back is more time- and energy efficient than turning on and off the full system.

The automation was rigorously tested in laboratory conditions as well as in real flight conditions on a test flight, as described in Section 3.1. The holographic system was successfully triggered by barometric altitude measurement and the holographic system automatically shut off after the disk was full. This automatic shutoff is essential to make handling of the instrument during landing easier and removes any danger from scattered laser light.

# 2.5 Laser Safety Considerations

The Laser used in the holographic system of the HoloTrack has laser Class IV. However, most of the pulse energy is absorbed within the optical system. For safety calculation we assumed a transmission of 

We performed experiments at two fan rotation rates, i.e. at two different mean velocities: 3.8 m/s and 10.0 m/s (current timing settings optimized for 10 m/s) with the turbulence grid open, meaning only acting as a passive grid. At the design velocity of 10.0 m/s we also increased the turbulence by operating the active grid and we tested the influence of a yaw angle on the validity of measurements in the holographic sample volume. In each of the experiments, droplets were introduced into the flow at the position of the grid with a hand held pressure sprayer and holograms were recorded with the timing as explained in Section 2.2.2 (hologram pairs at 25 Hz with inter-frame time of 500 µs). The recorded droplet sizes range from about 10 µm to 100 µm, but mostly >40 µm. From each hologram pair the droplet positions and size were extracted. The data recorded in the second frame "Holo  $H_1$ " are pre-shifted by the mean flow in y-direction (see Figure 10 for coordinate system) u measured with the pitot tubes as a first guess. Afterward, for both Holo  $H_1$  and Holo  $H_2$ , binary 2D images of the projected particle positions in the x-y plane are created.d By identifying the maximum of the two-dimensional correlation coefficient between the two images, the actual mean displacements in the x and y directions,  $\Delta \bar{s}_x$  and  $\Delta \bar{s}_y$ , are determined. The particle sizes in these projections are artificially enlarged, weighted by their square root, to enhance weight of small particles in the correlation. Within the overlapping region of the 18.4 mm × 18.4 mm center (blue and red square in Figure 10) regions of each holograms the particles are matched from Holo  $H_1$  to Holo  $H_2$ . For this, we search for matches within 500 $\mu$ m (pink square in top left of Figure 10 (a)) in x-y, 2 mm in z and an offset of 8  $\mu$ m or 20% of the diameter, which ever is lower. If more than one potential match is found, the closer match in position and size is selected. This simple matching procedure worked well for the sparsely populated Wind Tunnel Test Holograms but might need to be replaced with more sophisticated algorithms (e.g. Baek and Lee, 1996) or stricter rules for in-situ cloud droplet holograms.

## 3.3.1 Particle Match Rate and False Detection Rate

450

Before discussing the velocity measurements, we discuss the efficiency of droplet detection, which complements the CloudTarget results presented above. Through the matching, developed to analyze particle velocities, we can extract further information about how much we can trust the extracted particle data. From all the particles measured (i.e Predicted Positives) in the overlapping region in Hologram  $H_1$  ( $PP_{H1}$ ) a fraction can also be found in Hologram  $H_1$ , which we denote with  $PP_{H1 \wedge H2}$ . This ratio of particles that can be found in both holograms of a hologram pair to the total number of particles in one of the holograms we define as the Particle Match Rate PMR:

$$PMR_{H1} = \frac{PP_{H1 \wedge H2}}{PP_{H1}}. (6)$$

We calculate the PMR for 100 Hologram pairs of the wind tunnel test at two different velocities, so at two different shifts between holograms. Here, we use the data from experiments with lower turbulence where we used an open i.e. passive grid as we expect our simple matching algorithm to be even more reliable in less turbulent flows. In Figure 11, we show the PMR for different z-positions (positions of CloudTarget measurements  $\pm 1$ cm each) as a function of measured particle diameter  $d_m$ . We see a clear trend that match rate is both particle size and z-position dependent. This trend was expected as PMR is directly tied to recall.

Combining the results for Particle Match Rate with the recall measurements with the CloudTarget (see Section 3.2.1) allows us to determine the False Discovery Rate FDR and therefore a measure of False Positives FP. The idea follows the simple argument, that only real particles are matched from hologram to hologram as noise that produces FP would not be displaced with approximately the mean velocity which is a requirement for matching. If the recall is known a certain PMR can be expected. If the PMR is lower than this expected value, we assume False Positives to be the cause. We start with the definition of the Particle Match Rate and assume that there is no accidental matching, from which follows that all matched particles are True Positives  $PP_{H1 \land H2} = TP_{H1 \land H2}$ .

$$PMR_{H1} = \frac{PP_{H1 \wedge H2}}{PP_{H1}} = \frac{TP_{H1 \wedge H2}}{TP_{H1} + FP_{H1}} = \frac{1}{\frac{TP_{H1}}{TP_{H1 \wedge H2}} + \frac{FP_{H1}}{TP_{H1 \wedge H2}}}$$
(7)

We know that  $\operatorname{Recall}_{H1} = \frac{TP_{H1}}{P}$  and the probability, assuming the particle measurements in holograms  $H_1$  and  $H_2$  are completely independent, for particles to be found in both holograms  $H_1$  and  $H_2$  is  $\frac{TP_{H1 \land H2}}{P} = \operatorname{Recall}_{H1}^2$ . With that it directly follows

$$FDR_{H1} = \frac{FP_{H1}}{PP_{H1}} = 1 - \frac{PMR_{H1}}{\text{Recall}} \ . \tag{8}$$

The FDR (averaged over  $H_1$  and  $H_2$  of each hologram pair) is shown in Figure 11 on the right. The calculated FDR fluctuates around 0 for particles with  $d_m > 15 \mu m$  for all z, which indicates that there are almost no FP found. Negative values of FDR as observed for small particles and large z are not physical and therefore indicate an uncertainty in this method of evaluating the FDR. This uncertainty is a combination of uncertainty in particle matching PMR and the uncertainty in measuring the recall with the CloudTarget specifically for large z and small  $d_m$ . Especially for z = 17.4 cm we argue that the recall measurement

with CloudTarget probably underestimated the actual recall, which then leads to negative FDR, as it is unlikely that the matching used to calculate PMR was especially bad at high z. For smaller particles  $d_m < 12~\mu m$  the measurements become unreliable. This is indicated by a negative FDR for  $z=12.2~{\rm cm}$  and 17.4 cm. Moreover, less than 1.5% of the measured droplets had a diameter smaller than  $12\mu m$ , which translates to an average of less than 10 small droplets per hologram, so very few small False Positives FP (order of  $10^0$ ) or unmatched TP could lead to this overestimation of FDR here for small droplets in  $z < 10~{\rm cm}$  (FDR = FP/PP, low number of predicted positives PP in size bin means few FP could lead to high FDR).

## 3.3.2 Droplet Velocity Measurement Evaluation

From the one-to-one particle matching between holograms  $H_1$  and  $H_2$  the velocity of the individual particles can be calculated via  $u = -\frac{\Delta y}{\Delta t}$ ,  $w = -\frac{\Delta x}{\Delta t}$  where u is the oncoming flow velocity and w the vertical velocity. Due to high inaccuracies of measuring the z-positions of the particles ( $10^2 \, \mu m$ ) and the obstruction caused by the arms, the v component of the flow can not be accurately measured with the holographic system (see also Figure 13).

In Figure 12, we show the measured average particle velocity in the direction of the mean wind u from the holographic system normalized by the velocity measured by the 3D pitot tube. The mean measured particle velocities and mean velocity measured by the 3D pitot tube agree remarkably well within an offset of less than 3.5% for two different mean velocities. Moreover, the measured velocity is reasonably constant throughout the whole z-range between the holographic arms (z = 2.5 - 22 cm) with slightly higher measured velocities for small z as can be seen in Figure 12. For the lower velocity, we see that the standard deviation of the measured particle velocities (indicated with error-bars) exceeds the standard deviation from the pitot tube measurement (shaded region). At  $\bar{u} = 10$  m/s also the standard deviation agrees well. This is caused by the inter-frame time being  $500~\mu s$  in both cases, hence leading to a smaller displacement  $\Delta s_y$  in the lower velocity case. Since the absolute error in velocity is constant, the relative error becomes larger for lower mean velocity (see Figure 9). However, as explained earlier the error can be decreased by increasing the inter-frame time depending on the mean velocity.

To analyze HoloTrack's ability to measure fluctuating droplet velocities in a turbulent flow we compare results from experiments with an open grid to experiments with an active grid, increasing the turbulence in the flow. This analysis is restricted to the measurement volume that we propose as suitable in Section 4 to reliably detect droplets <10  $\mu$ m (1.84 cm  $\times$  2.34 cm  $\times$  5 cm with 3.5 cm<z <8.5 cm). In Figure 13 (a) and (b) we compare the probability density function (pdf) of the u-component of the 3D pitot tube with the pdf of the particle velocities measured with holography for a mean velocity of about 10 m/s with an open grid (A: lower turbulence) and active grid (B: higher turbulence). For the open i.e. passive grid the mean velocities agree well as discussed but the 3D pitot tube measures velocity fluctuations of  $u_{RMS} = \sqrt{\langle u' \rangle} = 0.16$  m/s, whereas HoloTrack measures higher  $u_{RMS} = 0.19$  m/s. The estimated uncertainty of the pitot tube pressure sensor is 0.05 m/s at 10 m/s which is 0.5% and HoloTrack's uncertainty is 0.07 m/s the deviation is therefore within the estimated uncertainties. In the higher turbulence case with the active grid, the 3D pitot tube and HoloTrack agree well with  $u_{RMS} = 0.38$  m/s (HoloTrack) and  $u_{RMS} = 0.38$  m/s (pitot tube), which confirms the accurate measurement of the fluctuating velocities further, especially when they are higher and the relative error decreases. The difference in measured mean velocity is only 1.2%. This can not be

exclusively explained by the pressure sensor uncertainty of the pitot tube (red shading). This slight offset could be caused by the two different measurement positions of holographic system and pitot tube and effects of the geometry of the HoloTrack instrument box that only cause a difference in the measured mean velocity in case of higher turbulence.

We have to keep in mind however, that the pitot tubes can also not be considered a perfect ground truth and there might be additional error sources besides the accuracy of the pressure sensors that can also shift the pitot tube results both for mean velocity as well as fluctuations.

In panels C and D of Figure 13, we show histograms of the velocity fluctuations in all three components. The grid configuration is expected to produce isotropic turbulence in the wind tunnel, so ideally all three components should agree. For the passive grid (Figure 13C), the fluctuations u' and v' agree remarkably well and  $u_{\rm RMS} \approx w_{\rm RMS}$ ; for the active grid (Figure 13 (d)), the agreement remains reasonable. The measured v'—the velocity fluctuations in the z-direction—however, exceed those of u' and u'. This behavior is expected, since measurements uncertainties associated with the v-component of the flow velocity is larger as discussed above. In contrast, the Pitot tube reports significantly lower fluctuations in the w component, with  $w_{RMS} = 0.24$  m/s for the active and  $w_{RMS} = 0.11$  m/s for the passive grid. Since Pitot accuracy decreases at low absolute velocities, also evident in a systematic offset for the mean velocity  $\bar{w}$ , its w-fluctuation values are less reliable. The comparison of u and w from HoloTrack is therefore likely more valid and supports the measured w component. The holographic system should have no systematic differences between x- or y- direction. Minor deviations could be caused by the instrument geometry affecting x- and y-velocity components differently. The observed negative mean vertical velocity, stronger in the less turbulent case, hints toward sedimentation of the droplets in the wind tunnel experiments. This is further attested by the settling velocity (-w) increasing with droplet diameter. Due to the limited data and to keep the scope of the paper clear, we refrain from further analyzing the relation in more detail here. Nonetheless, it showcases HoloTrack's useful feature of measuring droplet dynamics and size at the same time to investigate the dependence. Overall, we have seen that the velocity measurements of HoloTrack work as expected even with a very simple particle tracking algorithm. Mean velocity and velocity fluctuations that exceed the uncertainty in velocity measurement of 0.07 m/s, can be accurately measured in u (longitudinal) and w (vertical) direction.

## 3.3.3 Influence of Instrument Yaw on Measurement Accuracy

To analyze the effects of the arms on the holographic sample volume specifically in the case of non-zero yaw angle of attack we recorded holograms with HoloTrack being yawed with respect to the mean flow in the wind tunnel. These tests were performed with the grid open to have a close to laminar flow and see clear blockage effect of the arms on the flow. We investigate 4 different yaw angles:  $\alpha = 0^{\circ}, 1^{\circ}, 4^{\circ}$  and  $6^{\circ}$ . Here we define a positive yaw angle  $\alpha$ , when the flow has a negative v-component in z-direction as indicated in the schematic in Figure 14. We investigate positive yaw angles, as they are likely to have a stronger influence at the low z region of the sample volume which is more critical due to the higher recall for small droplets at low z. As a first indication of influence of the holographic arms, specifically the tips, we show a "super-hologram" i.e. a heatmap of relative concentration of detected droplets. In cases of optimal and constant detection and randomly distributed droplets we expect this heatmap to be flat. Any deviations indicate varying detection or a non-random particle distribution. In Figure 14

the super-hologram is shown as projection in x-z and y-z (where x was limited to the height of the arm tip, where the largest obstruction is) for the different yaw angles  $\alpha$ . In the case of no yaw we see that the region of <1 cm above the camera window shows lower particle concentration. This is caused by the boundary layer on the camera arm. For larger yaw angles we see that the height of the void region increases and the particles expelled from the wake of the arm accumulate in a distinct layer of high relative concentration. The angle of this accumulation layer in the x-z-plane can be associated with the angled tip of the camera arm, where the tip aligns with large x (bottom) of the camera. In the most extreme case of  $\alpha \approx 6^{\circ}$ , the void and accumulation regions reach up to  $z \approx 6$  cm. The other less significant non-uniformaties in the concentration further away from the camera observed in all yaw angles can be associated to the z-position and diameter dependent recall and non-random particle positions due to the hand-held spray bottle producing the droplets (each super-hologram is from data recorded within few seconds and spraying more towards low or high z can introduce a constant bias).

Another test, that is uniquely possible with HoloTrack is to analyze the influence of yaw angle of attack on the particle velocities. If the mean flow has a yaw angle of attack  $\alpha$  in the y-z-plane with respect to the y-axis of the instrument, in an optimal undisturbed case we expect the same angle  $\alpha_m = \alpha$  between the measured u and v component of the droplet velocities:

$$\alpha_m = \arctan\left(\frac{-v}{u}\right) \,. \tag{9}$$

In the previous section, we explicitly stated that the uncertainties in measuring droplet z-positions and therefore in measuring the v-velocity component of individual droplets is high. By averaging over droplets over several holograms and the whole x-y-domain, we are however able to see a clear signal and analyze the average droplet angle as a function of z-position, which is shown for the four different yaw angles in Figure 14. The observed  $\alpha_m \approx 0^\circ$  for  $\alpha = 0^\circ$  demonstrate the validity of this approach and that the high z-position- and therefore v-component uncertainty is averaged out by our approach and we do not have any persistent bias.

For all yaw angles >0°, the observed velocity angle  $\alpha_m$  in the center between the arms (z=12.25 cm) is approximately the yaw angle of attack  $\alpha$  and approaches 0° towards the arms. This means the flow aligns more with the direction of the holography arms the closer the z-position is to one of the arms.

We argue that quantities like concentration and size are largely unaffected by a slight deviation in velocity angle (change of  $10^{\circ}$  leads to change of <1 mm in position) but to accurately measure the droplet velocities and analyse clustering with e.g. the radial distribution function RDF of the droplets, where the accurate and undisturbed positions of droplets are of utmost importance, the analysis should be restricted to holograms with low yaw angle and z-regions of the sample volumes, where the measured angle droplet angle is undisturbed  $\alpha_m = \alpha$ . Our observation in turbulent wind tunnel flow, not shown here, indicate that the arm influence is less significant but we suggest the same restrictions as we found in the laminar case should be used to be on the safe side.

# 4 Discussion

## 4.1 Reliable Holography Sample Volumes for Droplet Position and Velocity Analysis

Table 2 quantifies the sampling volumes for two droplet sizes ( $d_m > 10 \, \mu m$  and  $d_m > 15 \, \mu m$ ) at a mean wind of 10 m/s, where the current inter-frame time of 500  $\mu$ s corresponds to a displacement of 5 mm. For particle positions and sizes, the combined volume of both holograms can be used, while for velocity only the overlap volume applies. For example, at 10 m/s, doubling the inter-frame timing from 500  $\mu$ s to 1000  $\mu$ s, thus increasing the mean displacement from 5 mm to 10 mm, decreases the overlap volume by 38%. The velocity RMS error is however reduced from 0.07 m/s to 0.03 m/s. This highlights the trade-off between velocity uncertainty and overlapped volume. Droplets with a minimal size of 10  $\mu$ m can be reliably detected up to  $z = 8.5 \, \text{cm}$ , and those larger than 15  $\mu$ m up to  $z = 12 \, \text{cm}$ , when restricted to the optimal  $18.4 \times 18.4 \, \text{mm}$  center in x-y. To avoid influence of the camera arm the sample volume should additionally be restricted to  $z > 3.5 \, \text{cm}$  at  $0^{\circ}$  yaw angle. The resulting reliable volumes are shown in Table 2.

Yaw angle during test flights was modest, with a standard deviation of  $6^{\circ}$ , confirming that the suspension design aligns the instrument with the mean flow. Wind tunnel experiments showed that even at optimal  $0^{\circ}$  yaw, valid measurements are restricted to z > 3.5 cm to avoid arm obstruction. Based on these constraints, the resulting reliable volumes are shown in Table 2.

For yaw angles  $|\alpha| > 1^\circ$ , concentration and velocity are affected up to z=6 cm or more. Therefore, for airborne high-precision velocity or position-sensitive analyses (e.g., turbulence fluctuations, RDF), only holograms with  $|\alpha| < 1^\circ$  provide sufficient data. But even in these the reliable volume is significantly reduced. For  $|\alpha| < 1^\circ$ , z > 5.5 cm ensures  $\alpha_m \approx \alpha$  and constant concentration, yielding  $1.84 \times 1.34 \times 3$  cm  $\approx 7.4$  cm³ per hologram. As an example, if the holograms in HoloTrack's maiden flight were not overexposed and could have been analyzed, this would have amount to  $\sim 34$  L of high-accuracy holographic data over a 20 min, 6.7 km transect (assuming only 15% of holograms are valid due to yaw angle). For comparison, a typical cloud droplet probe (e.g. CDP-2, Droplet Measurement Technologies) probe samples  $\sim 1.6$  L and the state-of-the-art MPCK+holographic system  $\sim 180$  L under ideal conditions. HoloTrack is unique, however, in providing simultaneous droplet imaging and velocity measurements, with two independent size/position estimates per particle.

# 4.2 Limitations and Potential Future Improvements

Future designs of HoloTrack could benefit from weight reduction and ideally the occasional USB disconnections that affect non-holographic sensors can be fixed. Increasing the disk space of the holographic system (currently RAID0 with 4× 1TB disks) can extend recording duration and total sample volume per flight. If new hard disks have even higher writing speeds, holograms could also be recorded in 10bit resolution, potentially enhancing detection. This potential effect of higher 10bit resolution on droplet detection can be evaluated with CloudTarget.

In wind tunnel tests, pitot tubes malfunctioned under heavy spray of large droplets but performed reliably in non-precipitating test flights, suggesting they only fail under extended exposure to precipitation-like conditions. HoloTrack's holograms were processed with the method of (Thiede et al., 2025b), achieving recall above 90% and negligible false positives for droplets down to 10 µin a defined sub-volume. The classification neural network is only trained on MPCK+flight data and even better

results may be achieved by fine-tuning the network with HoloTrack training data. The FDR can be estimated again for flight0 data to ensure in-situ-specific noise does not increase it significantly.

For in-situ applications, we plan to optimize sizing with the inverse threshold-independent method (Lu et al., 2012). Holo-Track's dual holograms per pair also allow an independent sizing uncertainty estimate. For higher turbulence and droplet concentrations than in our wind tunnel tests, more advanced particle matching algorithms (e.g. Baek and Lee, 1996) will likely be necessary.

Potential design refinements to mitigate the strong effect of yaw angle on the reliable hologram sub-volumes include: hanging HoloTrack with vertical z-axis (removing arm obstruction but risking window wetting in precipitation), stabilizing yaw with multi-point suspension, or optimizing 3D-printed arm tips. A key lesson is that "super-hologram" reconstructions alone cannot reveal all aerodynamic disturbances, especially in turbulence; only velocity calibrations as presented here can identify and correct these effects.

## 620 5 Conclusions

630

Overall, the evaluation confirms that HoloTrack is a powerful instrument for both laboratory and in-flight studies of cloud droplet microphysics. In the following we summarize the key points and results of evaluation tests:

- HoloTrack is a fully autonomous system that integrates a high-accuracy holographic sensor with environmental monitoring. It operates both in-flight and in laboratory settings with minimal operator input.
- The holographic system records 25 hologram pairs per second, with adjustable inter-frame time. At 500 μs, the in-plane velocity RMS error is less than 0.07 m/s while for the transversal component it ranges between 0.1-0.5 m/s depending on the z-position, but timing can be adapted to match expected wind speed and accuracy needs.
  - For particles down to 10 μm, recall exceeds 90% up to z = 8.5 cm, providing a reliable sample volume of 21.5 cm<sup>3</sup> (3D positions) or 12.3 cm<sup>3</sup> (3D velocities) per hologram pair. As an example, based on yaw measurements and defining reliable unobstructed sub-volumes, we estimate  $\sim$ 34 L of high-precision holographic data would have been usable of the holograms in HoloTrack's maiden flight (if the holograms were not overexposed due to a broken pinhole).
  - Longitudinal droplet velocity (u) agrees well with Pitot tube data, and isotropy of u and w fluctuations in grid turbulence further confirms the reliability of the velocity measurements, including small velocity fluctuations for in-plane components of u and w.
- By resolving both size and velocity of individual particles, HoloTrack enables studies of turbulence-induced particle interaction of spherical and non-spherical particles.

Code and data availability. Evaluation datasets and code are available from the authors upon reasonable request.

Author contributions. BT and GB designed and developed HoloTrack with help of EB. FN contributed the code for saving holograms to disks in real time. BT, YK and GB performed wind tunnel experiments. BT analyzed the holographic data from the performance evaluation
 experiments and YK calculated the dissipation rate from in-flight pitot tube measurement. All authors interpreted the results. BT wrote initial draft of the manuscript. All authors contributed to the final version of the manuscript.

Competing interests. The authors declare no competing interests.

645

Acknowledgements. We thank the MPI-DS machine shop, led by Andre Heil, for designing and manufacturing HoloTrack's mechanical setup. We also extend our gratitude to the MPI-DS research electronics team, particularly Laura Diaz-Maue, for designing the electronic components of the communication and power systems, and Holger Nobach, for advising on laser safety and for the development of the sequence generator. We thank Venecia Chavez-Medina and Hossein Khodamoradi for providing codes and advice for the use of the environmental sensors. Many thanks to the entire IMPACT campaign team for enabling the success of the campaign and the test flight.

This work was partly supported by the Fraunhofer–Max Planck cooperation program through the TWISTER project. Birte Thiede was financially supported by a fellowship of the IMPRS for Physics of Biological and Complex Systems.

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

| Instrument Name                       | Manufacturer                   | Measured Quantities                                                                                                                                     | Nom. Acquisition Rate [Hz]                       |
|---------------------------------------|--------------------------------|---------------------------------------------------------------------------------------------------------------------------------------------------------|--------------------------------------------------|
| Holographic System                    | in-house                       | individual 3D particle position (21.5 cm³ per pair)<br>cross-section size and shape (21.5 cm³ per pair)<br>2D particle velocity u,w (12.3 cm³ per pair) | 25 (hologram pairs) 50 (individual holograms)    |
| VectoDAQ                              | Vectoflow GmbH                 | 3D flow velocity $u,v,w$                                                                                                                                | 50                                               |
| PSS8 ADC                              | Simtec AG                      | 1D flow velocity $u$                                                                                                                                    | 100                                              |
| SBG Ellipse-N                         | SBG Systems                    | 3D orientation, velocity, and GPS position                                                                                                              | Acc. 390, Gyro. 133,<br>Magn. 22, GPS 5          |
| OPC-N3                                | Alphasense                     | Particles, 0.35μm to 40 μm                                                                                                                              | 1                                                |
| SHT40                                 | Sensirion                      | Temperature, Relative Humidity (RH)                                                                                                                     | 15.3                                             |
| BMP390                                | Bosch                          | Temperature, Pressure                                                                                                                                   | 15.3                                             |
| TMP117                                | Texas Instruments              | Temperature                                                                                                                                             | 15.3                                             |
| BME688                                | Bosch                          | Temperature, RH, absolute pressure, trace gases                                                                                                         | 1                                                |
| <b>Future Instrumentation</b>         |                                |                                                                                                                                                         |                                                  |
| simpleRTK2B with 3x<br>U-Blox ZED-F9P | ardusimple, U-Blox             | GPS Data, 3D orientation                                                                                                                                | 10                                               |
| CDP-2                                 | Droplet Measurement Techniques | Particles in quasi 1D, 2 μm to 50 μm                                                                                                                    | continuous in 0.24 mm <sup>2</sup> cross section |

**Table 1.** Overview about the different measurement systems combined in HoloTrack. The main system is the holographic setup, supported by measurement of instrument position and movement as well as flow properties and measured quantities like temperature and relative humidity. The OPC-N3 and CDP-2 are additional particle sensors.

|                            | $d_m > 10~\mu\mathrm{m}$                                                         | $d_m > 15~\mu\mathrm{m}$                                                           |
|----------------------------|----------------------------------------------------------------------------------|------------------------------------------------------------------------------------|
| Particle Position and Size | $1.84 \text{ cm} \times 2.34 \text{ cm} \times 5 \text{ cm} = 21.5 \text{ cm}^3$ | $1.84 \text{ cm} \times 2.34 \text{ cm} \times 8.5 \text{ cm} = 36.6 \text{ cm}^3$ |
| Particle Velocity          | $1.84 \text{ cm} \times 1.34 \text{ cm} \times 5 \text{ cm} = 12.3 \text{ cm}^3$ | $1.84 \text{ cm} \times 1.34 \text{ cm} \times 8.5 \text{ cm} = 21.0 \text{ cm}^3$ |

Table 2. Holographic sample volumes per hologram pair if the mean velocity leads to a displacement of 5 mm (e.g. current inter-frame time of 500  $\mu$ s with a mean wind speed of 10m/s) at an angle of attack of 0°. To capture particle positions and their size the combined volume of both holograms can be used, for velocity measurement only the overlapping region is considered. We show volumes for two different minimal droplet diameters, where recall >90%. Multiplying values of the sample volume by 25 Hz, i.e. the HoloTrack double-frame acquisition frequency, provides the sampling volume per seconds.

Figure 1. HoloTrack's design concept. Schematic showing HoloTrack's camera and laser arms and the sampling volume in droplet-laden turbulent flow (left), and a pair of hypothetical holograms of droplets,  $H_1$  and  $H_2$ , with overlapping sample volumes (right). With a short enough inter-frame time  $\Delta t$  the particles on the upstream part of sample  $H_1$ , shown in blue, are captured in the downstream part of  $H_2$ . By matching droplet D between hologram  $H_1$  and  $H_2$  the 3D displacement and, hence, the 3D velocity can be calculated with  $\mathbf{u} = (w, u, v) = (\bar{w} + w', \bar{u} + u', \bar{v} + v') = ((x, y, z)_{DH_1} - (x, y, z)_{DH_2})/(\Delta t)$ , where an overbar denotes the mean (averaged) component and a prime denotes the fluctuating component. The droplets that are captured in both holograms of the pair, and for which the velocity can be calculated accordingly, are shown in blue. For the remaining droplets of each hologram, shown in green here, only size and position is known and velocity can not be determined.

Figure 2. HoloTrack is an instrument box primarily designed for in-situ measurement of cloud droplet dynamics on the Max-Planck-CloudKites, while also being optimized for simple deployment in laboratory measurements. The top-right panel shows HoloTrack during its maiden flight aboard a CloudKite; the other panels present multi-view CAD visualizations of the instrument. The dimensions of the instrument are marked in the middle left panel, which is a top view of the instrument design plan. The instrument consists of the main box including electronics and devices for measurement control and acquisition. The arms contain the holographic system with camera and laser beam path. The measurement status of HoloTrack can be observed via a screen on top of the instrument. The holographic sample volume is shown in green. 1D and 3D pitot tubes are installed in the direction of the flow. In the cap small-scale sensors to measure environmental quantities and the optical particle counter are installed. For in-flight measurements a battery and a stabilizer fin can be fixed to the back of the instrument and landing feet ensure the sensitive parts of the instrument are always far from the ground in field measurements.

**Figure 3.** Beam path for laser beam alignment, expansion and collimation. In the top panel, the actual construction within the laser arm is shown. Optical elements are fixed in optical mounts, which are further stabilized (through the Thorlabs Cage System). An adjustable mirror aligns the laser beam into the laser arm. The first two lenses for collimation are placed in x-y-translational stations, and the pinhole for spatial filtering is positioned at the beam waist in the focus of the first lens with the help of a x-y-z-translational stage. Behind the second lens, the beam intensity is reduced with a neutral density filter and the beam diameter is reduced with a circular aperture. The third lens is used to further expand the beam. The final lens collimates the beam and is therefore movable in z-direction. All holders are fixed with several screws into the base plate and/or stabilized by metal rods for optimized alignment. The bottom panel shows a simulation of the expected beam diameter as a function of z-distance, and collimation within the limits of available aspheric lenses, lens diameters and overall length of the beam path.

**Figure 4.** Timing diagram for recording one hologram pair with effective inter-frame time of 500µs. The first camera exposure is 100µs long and right before the camera shutter closes the first laser beam of the laser pulse burst is emitted. The burst consists of 41 pulses with a frequency of 80 kHz. Between the camera exposures 39 laser pulses are not recorded. The 41st laser pulse is right in the beginning of the long second exposure of the second hologram per pair. The longer exposure is limited to the read out time of the first hologram. The second exposure, is however effectively reduced to about 100µs with the help of a fast liquid crystal shutter.

**Figure 5.** A: Mounting on HoloTrack on the MPCK platform with a 3 m line from the keel. B: HoloTrack in flight on the MPCK platform. The red arrow shows the HoloTrack hanging 6 m below the lower Helikite of the MPCK platform. C: Overview about test flight. The holographic system was running in altitude trigger mode with a limit of 700m of barometric altitude. The yellow point indicates when the control system is turning the holographic system on. Shortly after the holographic system starts acquiring images for about 20 minutes until the disks are full (4TB at 25 hologram pairs per second). The system successfully shut off when altitude was below the limit again.

**Figure 6.** From the SBG the motion in terms of Euler Angles of the HoloTrack during flight are analyzed. Here, we show the motion for barometric altitude >700 m, which is the altitude chosen for holographic measurements in the test flight. The mean yaw angle changes with altitude and the fluctuations are on the order of  $10^{\circ}$ . From the 3D-velocity measurements the flow angles reveal that HoloTrack aligns well with the mean flow (mean yaw angle close to 0). Pitch angle shows influence of relative vertical velocity due to upward/downward motion.

Figure 7. (a): Velocity fluctuation measured with 1D (100 Hz, 8-point averaging effectively 12.5 Hz) and 3D pitot tubes (50 Hz) in a region of  $\approx$  820 m altitude show overall agreement. Sections with continuous measurements are marked with shading. (b): The second order longitudinal structure function only reveals  $r^{2/3}$ -scaling in the inertial sub-range for the measurements with 3D pitot tube. (c): From the structure function the dissipation rate  $\varepsilon$  was determined for the shaded red region shown in panel (a) (820 m) and another region at lower altitude (570 m).

Figure 8. Recall measured with the CloudTarget as a function of the theoretical ground truth particle diameter  $d_{gt}$ . The recall is determined within the center cross section of to  $18.4 \times 18.4$  mm. A CloudTarget photomask was recorded with HoloTrack at different z-distances from the imaging plane. The holograms were automatically processed. Recall is a measure for detection efficiency and indicates how many of the actual particles were correctly found by the system. For  $z \lesssim 8.5$  cm particles of  $10 \, \mu m$  diameter or larger are reliably detected. The results are binned by ground truth diameter  $d_{gt}$ , the diameter of the printed disks on the photomask which might vary slightly from the measured diameter  $d_m$ .

**Figure 9.** CloudTarget reveals error in inter-particle distance measurements. Histograms of inter-particle distance error are shown as a function of sample volume depth z. The RMS distance error remains below 33  $\mu$ m in all cases, corresponding to an upper-bound velocity error of 0.07 m/s for an inter-frame time of 500  $\mu$ s. As before, the analysis cross-section is limited to the central 18.4  $\times$  18.4 mm region in x-y.

Figure 10. Left: HoloTrack placed in Wind Tunnel for evaluation of particle tracking. The y-axis of the sample volume is aligned with the mean flow direction u in the non-yawed experiments. The sample volume is 19 cm above ground. Right: Examples of particles measured in a hologram pair, that consists of hologram  $H_1$  and  $H_2$ . For each hologram the center x-y cross section of  $18.4 \times 18.4$  mm is considered (shown as red and blue square) and matching is performed in the overlapping region. Particles that are considered a match are marked with a dark red outline and need to be within  $500\mu$ m in x-y after mean shift (indicated by small pink square) and within 2mm in z to each other, and can only deviate 20% (or  $8\mu$ m) in diameter to be considered a match.

Figure 11. Left: Particle Match Rate as a function of measured diameter for the same z-positions used in the CloudTarget Test (each z corresponds to  $z\pm 1$ cm). The Match Rate is calculated based on the overlapping cross sectional regions of  $18.4\times18.4$  mm and is a measure for how many of the measured particles are found in both hologram  $H_1$  and  $H_2$ . Right: Taking the recall determined with CloudTarget into account allows an estimation of FDR, which independently of z-position is negligible for particles larger than  $15\mu$ m. For smaller particles, the total number of sampled particles were too low in the Wind Tunnel tests to draw reliable conclusions.

**Figure 12.** Particle velocity measured with HoloTrack as a function of position between the arms normalized with the mean velocity measured by the 1D pitot tube. Errorbars indicate the standard deviation of the measured droplet velocities from the holographic system and the shaded region indicated the standard deviation of velocities measured with the 3D pitot tube. The offset of the mean is smaller than 5% which shows the arms only have minimal effect on the flow if HoloTrack is directly oriented into the mean wind.

Figure 13. Comparison of longitudinal 3D-pitot-tube and holographic (HoloTrack) u velocity measurements for passive (A) and active (B) grid conditions and comparison of velocity fluctuations in all three directions measured with HoloTrack for passive (C) and active (D) grid conditions. Accurate measurement of longitudinal component u is confirmed by comparison to 3D-pitot measurement (A,B), while  $w_{RMS} \approx u_{RMS}$  in isotropic conditions confirms accurate measurement of w, v' is biased by high z-position uncertainty. Panel E and F show diameter dependent vertical droplet velocity w to showcase HoloTrack's strength in measuring size and velocity simultaneously.

Figure 14. If the mean flow has a non-zero yaw angle with respect to HoloTrack's y-axis the holographic sample volume is influences by the obstructing arms. The influence can be analyzed with super-holograms revealing void regions and regions of droplet accumulation. The holographic arms also force the flow to align with the direction if the arms, which is revealed by analyzing the z-dependence of the droplet velocity direction angles  $\alpha_m$  that approaches  $0^\circ$  in the vicinity of the arms. If  $\alpha \neq 0$ , which can reliably measured with the 3D pitot tube, the usable sub-volume of the holographic sample volume needs to be adjusted.