# Peer review of "HoloTrack: In-Situ Holographic Particle Tracking Velocimetry of Cloud Droplets"

_EGUsphere, 2025_

## Author Comment (AC1)

Reply to Anonymous Referee #1

August 29,2025

Dear Reviewer,

We sincerely thank you for your thoughtful and constructive review of our manuscript "HoloTrack: In-Situ Holographic Particle Tracking of Cloud Droplets." We greatly appreciate the positive assessment of the instrument's design, evaluation, and potential impact on cloud microphysics research. The comments and suggestions helped us further improve the clarity and utility of the paper. Below, we address each comment in detail one by one. Below, reviewer comments and question appear in blue and our responses are shown in black.

**Review of "HoloTrack: In-Situ Holographic Particle Tracking of Cloud Droplets"**

**Overview of the Paper:** *This manuscript presents HoloTrack, a novel, fully autonomous instrument for the in-situ measurement of cloud microphysical properties. The primary innovation of HoloTrack is its ability to perform three-dimensional particle tracking by capturing pairs of holograms at a high frequency (25 pairs/sec). This allows for the direct measurement of individual cloud droplet velocities in addition to their 3D position, size, and shape. The authors provide a comprehensive description of the instrument's mechanical, optical, and electronic design, as well as its automation systems. The paper's strength lies in its thorough performance evaluation, which includes a maiden test flight on the Max Planck CloudKite (MPCK) platform, static tests using a calibrated "CloudTarget," and a series of detailed wind tunnel experiments. These evaluations quantify the instrument's detection efficiency, velocity measurement accuracy and uncertainty, and the aerodynamic influence of the instrument's body on the sample volume under various yaw angles. The work represents a significant technical achievement and provides a powerful new tool for advancing the experimental understanding of cloud microphysics, turbulence, and droplet dynamics.*

**General Recommendation:** *The paper is well-written, the instrument is thoughtfully designed, and the performance evaluation is extensive and convincing. It is a substantial contribution to the field of atmospheric measurement technology. The conclusions are well-supported by the presented data. The manuscript is nearly ready for publication.*

*I recommend this paper for publication after minor revisions. The revisions suggested below are intended to clarify a few points, which will enhance the paper's impact and utility for future users of this technology.*

**Comments:**

*(C1): Table 1 (page 10): Two instruments are listed under the heading "Planned but not operational." To enhance clarity, consider naming the subsection: "Future Instrumentation"—to clearly indicate that they were not part of the current test configuration.*

(A1): We thank the reviewer for this helpful suggestion to improve clarity. We have renamed the subsection of the table to "Future Instrumentation" according to your suggestion to indicate these instruments were not part of the test configuration.

*(C2): Line 111: For unambiguous interpretation, please state explicitly which side of the windows the measurements refer to (e.g., "interior-facing" or "exterior-facing").*

(A2): We thank the reviewer for pointing out this ambiguity. We refer to the exterior-facing sides of the windows. We have reworded the sentence to "The exterior-facing side of the camera window is at reconstructed z = 2.5cm and the exterior-facing side of the laser window at z = 22cm." to clearly indicate that anything between 2.5cm and 22cm can be considered the actual sample volume.

*(C3): Line 165: For clarity, consider labeling the holograms as $H_1$ and $H_2$ instead of A and B. The use of A/B within the sentence structure may cause confusion.*

(A3): We thank the reviewer for this helpful suggestion. We have replaced the A/B labels with $H_1$ and $H_2$ not only in line 165 but for consistency also throughout the manuscript including legends of Figures and already introduce this naming convention in the introduction and the new Figure 1 for clarity.

*(C4): Line 170: Please clarify why the second exposure (B) must be exactly 14 ms. Is it not possible to wait 13.9 ms and then use an exposure time of 0.1 ms instead?*

(A4): We thank the reviewer for pointing out this ambiguity. We have revised the text to clarify that the second exposure H2 must be $t_{H2} \approx t_{rd} - \Delta t$, where $\Delta t$ is the time interval between the two exposures. For particle tracking velocimetry, $\Delta t$ is typically small compared to $t_{rd}$, making $t_{H2} \approx t_{rd} \approx 14$ms a good approximation. This revision makes the statement both more correct and complete.

*(C5): Figure 6: For consistency and easier referencing, use (a), (b), (c) to label the panels.*

(A5): Following the reviewer's suggestion we have added A,B,C to the panels in Figure 6 and added it wherever panels of Figure 6 are referred to in the text.

*(C6): Line 275: Please specify the distance between the instrument and the balloon. Additionally, address whether the balloon has any potential influence on the measurements (e.g., wake effects, thermal interference, shadowing)*

(A6): We have added the information about the distance between balloon and instrument as well as information about the intended mounting option for future flight in line 289 of the revised manuscript: "For the maiden flight, HoloTrack was attached with a line about 3 m below the keel of the lower Helikite, which results in a distance of 6 m to the balloon for ease of operation (see Figure 5 A). The hanging point is approximately 6.5 m downstream of the balloon edge. Hence, the wake does not reach HoloTrack as long as the pitch angle of attack is smaller than approximately 45°. As demonstrated in Figure 6, in the test flight the standard deviation of the pitch angle was only 10° with a mean of 0°. While even in this configuration the effect of the balloon is minimal, in future flights HoloTrack can be hung on a line directly from the tether at arbitrary distances below the balloon e.g. 10 m to 1000 m (as shown for WinDarts in Chávez-Medina et al., 2025).
In addition, we want to note that previous measurements with a different instrument (Advanced Max Planck CloudKite or MPCK⁺) have shown comparable turbulence dissipation rates for both belly and keel mounting (directly below the balloon and 3 m below), indicating that turbulence is likely not affected by the balloon even when the measuring instrument is closer to it than in the HoloTrack maiden flight. Dynamic pressure field distortions, however, can influence the mean velocity, which is why future deployments will consistently use a tether mount further away from the balloon than in the maiden flight, as now indicated in the text. Potential effects of the Balloon (Advanced Max Planck CloudKite platform) and different mounting options will be discussed in a future publication, currently in preparation, about the Advanced Max Planck CloudKite system (Helikites and instrument box). Importantly, the maiden flight configuration was still valuable because the instrument was freely suspended and therefore experienced motion and stability conditions similar to those expected in the final tether-mounted configuration. Thermal effects from the balloon will also be negligible in tether-mount configuration.

*(C7): Line 298: Typographical correction: replace "near 0°" with the correct form "near 0°"*

(A7): We thank the reviewer for pointing this out. The typographical error has been corrected to "near 0°".

*(C8): Line 316: 1D Pitot Tube Filtering: The text notes that an "8-point-filtering was still set" on the 1D pitot tube, which smoothed the data. To aid reader understanding, please clarify why this filtering was active during the test flight—was it due to a default configuration, an oversight, or intentional for noise reduction?*

(A8): We thank the reviewer for this comment. The filtering was part of the instruments default configuration. We now refer to it throughout the manuscript as "default 8-point filtering" to make this clear.

We want to thank the reviewer again for their helpful comments and positive assesment. We have made additional minor revisions to streamline the manuscript and enhance clarity. Furthermore, the CloudTarget analysis for $z = 12.2$ has been improved with a more accurate tilt and "ghost-layer" (see Thiede 24, EGUsphere, 2025, 1–39, 2025b) correction. The velocity results in Figure 13 (previous Figure 12) are now shown for the final sub-volume, for consistency, which is also indicated in the revised manuscript. All other revisions are clearly indicated in the marked-up version of the manuscript.

---

## Author Comment (AC2)

**Reply to Anonymous Referee #2**

**August 29,2025**

Dear Reviewer,

we sincerely thank you for your careful reading of our manuscript and for your constructive and detailed feedback. Your comments were very helpful and have allowed us to clarify and improve the structure and readability of the paper. We believe that addressing your suggestions has significantly strengthened the manuscript. In the following, we provide a point-by-point response, with the reviewer's comments shown in *blue* and our replies in black.

*This paper thoroughly and rigorously describes an instrument for tracking of cloud droplets measured by a holographic imaging system. The paper should be accepted after addressing the following points:*

*Major comments:*

*(C1): The paper is very detailed, even tedious in places. Any streamlining would be appreciated, especially to reduce repetition. In particular, much of the material in the "Discussion" section is not really discussion, but simply repeating what has already been stated clearly in the main text of the paper. The Discussion section should be rewritten to focus on putting results into the bigger scientific context and describing plans for future improvements or applications.*

(A1): Thank you for this important point, which helped us improve the focus of the manuscript. We have restructured the Discussion section into two parts: 4.1 "Reliable Holography Sample Volumes for Droplet Position and Velocity Analysis" describing which sample volumes can be trusted based on the desired scientific analysis and yaw angle, an important clarification to establish for future applications, and 4.2 "Limitations and Potential Future Improvements". We carefully revised the text to avoid repetition where no new information was added in the discussion part. Overall, we believe this has improved the overall clarity and structure.

*(C2): The big picture is missing, especially for those who may not be familiar with the concept of PIV or particle tracking. In the Introduction the measurement approach should be more clearly explained: two holograms recorded in rapid succession, such that they have significant spatial overlap, and then matching particles observed in both sub-volumes in order to obtain a velocity vector. One part of that vector is due to the mean advection of the flow, the other part is due to turbulence. Perhaps a figure illustrating this, the way you would explain in a presentation to a broad audience, could be useful.*

(A2): We thank the reviewer for pointing this out. To make the general approach and idea of the holographic system in HoloTrack for velocity measurement clearer, we have followed the reviewer's suggestion and added a schematic figure to the introduction. We further describe the approach with "We have developed HoloTrack, the first holographic cloud droplet tracking velocimetry instrument that simultaneously measures droplet size and 3D velocity in a large, localized sample volume. By combining the low true-air-speed of the Max Planck CloudKite platform with advanced camera technology and precise timing, HoloTrack captures hologram pairs with overlapping volumes, enabling droplet tracking in three dimensions. Figure 1 illustrates the principle: the system records a hologram pair ($H_1$ and $H_2$), and, based on the inter-frame time dt and mean true-air velocity u, droplets captured in the upstream part of $H_1$ are also captured downstream in $H_2$ (blue droplets in Figure 1). Matching droplets across the hologram pair allows computation of 3D velocities, though the z-component is associated with higher uncertainties." in line 55 of the revised manuscript.

(C3): On this point: is this really "tracking" or is it velocimetry? With only two frames, it seems like a stretch to refer to this as particle tracking. But I agree it's also not traditional PIV based on image correlation. Is there standard terminology in the field? If so, it's best to introduce it and clearly define it.

(A3): We thank the reviewer for this important point regarding terminology. While to our knowledge the terminology is not unambiguous, we now use particle velocimetry instead of particle tracking throughout the manuscript and in the title as it does not necessarily imply a continuous tracking of particles. We clarify our approach even further by calling our approach two-frame particle velocimetry in the abstract (in line 11 of the revised manuscript).

(C4): In the end it's not clear whether holograms were obtained from within natural clouds. On line 268 it is stated that in-situ holograms could not be evaluated, but the reason is not clearly stated. If no in situ data are available, it should be stated directly that the tests of the holographic tracking are under laboratory conditions only. This will ensure the scope of the paper is accurately stated.

(A4): We thank the reviewer for this suggestion, we did, however, clearly stated the reason why in-situ holograms could not be evaluated in line 268 of original manuscript: "During the IMPACT campaign (May-June 2025, Pallas Finland), HoloTrack had its maiden flight, successfully collecting various datasets, including holograms, as planned. **Although a broken pinhole in the holographic optical system rendered the collected holograms too bright to be usable**, the test flight still demonstrated HoloTrack's ability to operate effectively under flight conditions."
We have made further efforts to more clearly state the scope of our tests and evaluation by improving the beginning of Section 3 ("Performance Evaluation"). We have included "**After replacement of the pinhole, evaluation of the holographic system was carried with experiments under laboratory conditions.**"
Additionally in the abstract we now clearly state: "**With laboratory tests we confirmed** , that the holographic system reliably detects particles down to 10 µm, within a sample volume of 17 cm$^3$ (1.84 cm ×1.84 cm ×5 cm) of each hologram." and "A series of laboratory tests and a maiden flight tests…"

*Detailed comments:*

(C5): Lines 8-9: It is not clear in the abstract where these volumes come from and what they mean. If you wish to keep the numbers, please explain why the volumes are different, i.e., that the velocimetry depends on two samples separated by very short time, and therefore overlapping in space.

(A5): We have rephrased that section to "With laboratory tests we confirmed, that the holographic system reliably detects particles down to 10 µm, within a sample volume of 17 cm$^3$ (1.84 cm ×1.84 cm ×5 cm) of each hologram. For a recorded hologram pair with mean displacement of 0.5 cm caused by e.g. an inter-frame time of 500 µs and a mean velocity of 10 m/s, this results in 21.5 cm$^3$ combined volume, where particle position and size is sampled and 12.3 cm$^3$ overlapping volume where the two-frame particle velocimetry can be applied to resolve individual droplet velocities. Reliable sub-volumes for measuring droplets at different yaw angles, to account for the influence of the instrument body are further defined." to explain in more detail that the volumes are because of limited detection and overlapping and combined volumes.

(C6): Line 19: It's an oversimplification to state that cloud properties are determined by the microphysics of clouds. One could just as easily claim that microphysics and other cloud properties are determined by cloud dynamics. The truth is that microphysics both follow and help determine other large-scale cloud properties. Please rephrase.

(A6): We thank the reviewer and agree that this statement was indeed an oversimplification. We rephrased this sentence to "Cloud properties emerge from a complex interplay between microphysical processes, such as droplet size, distribution, and dynamics and large-scale cloud dynamics.

Microphysics both respond to and influence the thermodynamic environment and turbulent motions within clouds (Shaw, 2003) with droplet size evolution closely linked to turbulent flow and the history of entrainment and mixing (Grabowski and Wang, 2013)."

(C7): Line 29: It seems strange to cite a review paper on mixed-phase clouds as a reference for holographic cloud imaging. Perhaps there is a more appropriate one?

(A7): Thank you for pointing this out. The reference was intended to provide an overview of different types of optical probes rather than specifically for holographic cloud imaging. We agree the placement was confusing and have revised the sentence accordingly. The references now appear in the context of the general division of probes: "Generally these can be divided into two groups (see discussions about probes in e.g. Beals, 2013; Korolev et al., 2017): traditional probes measuring a single particle at a time, probing a quasi-1D volume and camera based measurement…".

(C8): Line 43: The sentence would read better with a comma after "cloud droplet measurement".

(A8): We have added the comma accordingly.

(C9): Last two paragraphs of Section 1: I was confused until the very end, and in fact I thought the paper was about a 2D PIV system. It takes too long to figure out what the paper is about, and even in the last paragraph it's not really clear how the system works (see related point in "Major Comments" above). For example, "first airborne instrument capable of capturing hologram pair tracking of droplets" is quite obtuse. Please restructure the Introduction to clearly state the problem and to clearly outline the overall approach so readers can have an idea of what the paper is about.

(A9): We thank the reviewer for raising this important point. We have improved the introduction, specifically by shortening the general introductory part about cloud microphysics and measurements. We clearly state the problem in line 41 of the revised manuscript: "Despite the described advantage of these measurements and recent achievements of holographic cloud droplet measurements, a key aspect of cloud microphysics remains largely inaccessible: droplet dynamics." Regarding the overall approach we have added a figure and text introducing this, see our reply to comment C2.

(C10): Line 78: Delete "at most".

(A10): We have deleted "at most" accordingly.

(C11): Caption for Figure 1: I don't know what OPC-N3 is referring to… need to define.

(A11): We thank the reviewer for pointing this out. We have replaced OPC-N3 with "optical particle counter" in the caption of Figure 1 with: "In the cap small-scale sensors to measure environmental quantities and the optical particle counter are installed".

(C12): Line 111: The concept of reconsruction has not been introduced, so please do that first.

(A12): Whereas, in the original manuscript in line 101 "wavefront reconstruction" was mentioned, the revised manuscript now states more clearly "…with wavefront reconstruction via the Huygen-Fresnel kernel (Fugal et al., 2009)." to introduce the concept more specifically.

(C13): Line 112: Is 8-bit resolution sufficient to resolve the fringe visibility needed for high spatial localization of particles and particle size? See, e.g., discussion of fringe visibility in Fugal et al. 2004 (Applied Optics).

(A13): We believe that a larger bit depth could indeed be beneficial for increasing signal, although in in-situ operation the noise level (e.g., from contaminated windows, remaining diffraction patterns from particles out of focus) is already quite strong. Moving to 10-bit resolution would actually not increase readout time or reduce frame rate in HoloTrack's camera, therefore our timing scheme would be unaffected. At present, the limiting factor is the hard disk RAID write speed, which restricts operation to 50 fps rather than the maximum 71 fps in 8-bit operation already. For 10-bit, a reliable writing to our current disks would reduce the usable framerate to 38fps or 19 hologram pairs per second. Going to 12-bit would halve the frame rate of the camera due to a doubled readout time.

It would certainly be interesting to repeat cloud-target tests with higher bit depth to evaluate changes in recall, particularly for small and distant particles. In the end, this remains a design decision, balancing potential increase in signal strength with potential decrease in frame rate translating to an increase in spatial inter-sample distances. To acknowledge this, we have added the following to the discussion Section 4.2: "If new hard disks have even higher writing speeds, holograms could also be recorded in 10bit resolution, potentially enhancing detection. This potential effect of higher 10bit resolution on droplet detection can be evaluated with CloudTarget."

*(C14): Line 126: State "Section 3.2".*

(A14): We thank the reviewer for this suggestion and have made the change accordingly.

*(C15): Line 130: Show numbers to convince that this limit is not typically reached. I believe that it is reached in polluted clouds.*

(A15): Thank you for this suggestion. We agree that in polluted clouds the stated limit can be reached. The mentioned limit itself is an estimate, and we now provide more detail, including two references on shadow-density constraints (see line 133 of the revised manuscript). We also added numerical examples for a high droplet number concentration of 500 cm$^{-3}$ across representative droplet sizes, showing that, given the current optical path with a 19.5 cm separation, the projected area fraction (shadow density) typically remains below or on the order of the cited limits. We note explicitly that larger droplets at high concentration may still degrade hologram quality.
"Additionally, the shadow density increases with larger z-component of the sample volume. According to empirical results by Royer (1974) hologram quality deteriorates for SD > 1% and a theoretical upper bound limit for which holography becomes unsuitable is given by Meng et al. (1993) with approximately 40% (G = 1 in Meng et al., 1993). With a 19.5cm z-extent, the shadow density is SD = $\pi$19.5cm/4 sum($n\_id\_i^2$) which would be 1%, 3%, 7% for 500 cm$^{-3}$ monodisperse droplets of 10,20,30 $\mu$m respectively. Hologram quality is therefore expected to only be strongly affected in conditions with exceptionally large number concentration and droplet sizes".

*(C16): Line 132: Three uses of "sample" "sampled" and "sampling" in one sentence… please rephrase to make it clearer and less repetitive.*

(A16): We appreciates the reviewer pointing this out and have rephrased the sentence to "At a nominal mean velocity of 10 m/s this yields a three-dimensional sample every 40 cm horizontal distance resolving the cloud at sub-m resolution."

*(C17): Line 139: I don't understand where the 45 cm constraint comes from… please explain more clearly.*

(A17): We rephrased to "In HoloTrack this expansion has to be achieved over a beam path of approximately 45 cm based on the current design setup." to make clear the constraint comes exclusively from our design choices.

*(C18): Figure 2 caption: Meaning of "Thorlabs systems" is unclear. Also, meaning of "this simulation code" is unclear.*

(A18): We have rephrased the mentioned sentences in the caption to "Optical elements are fixed in optical mounts, which are further stabilized (through the Thorlabs Cage System)." And "This simple simulation made with lens equations was used to optimize expansion" to more clearly state it is a self-made simulation with a simple code that takes lens equation into account.

*(C19): Line 182: "Despite the simplicity of the described timing."*

(A19): We thank the reviewer for pointing us to this phrase and have rephrased the sentence to "The triggering signals emitted by the sequence generator have to take the laser, shutter and camera delays into account and the LC-shutter requires a specific signal pattern to be in the open or closed state.".

*(C20): Section 2.2.2: It's not clear why the laser is run at such a high repetition rate. Why not simply fire two pulses with the required delay? Presumably this must not be possible, but the reason is unclear.*

(A20): We thank the reviewer for the comment. While it is in principle possible to operate the laser at lower frequencies, the optimum pulse energy is reached at 80 kHz, and at lower repetition rates the beam intensity may not be sufficient depending on the desired inter-frame time.
The primary reason for operating at 80 kHz is to maintain flexibility in timing and, consequently, the effective inter-frame time. The output power varies with the laser repetition rate, so neutral density filters in the laser beam path would need to be exchanged at lower frequencies to maintain hologram brightness. Additionally, the laser includes a first-pulse suppression feature, which ensures that the typically brighter first pulse matches the energy of subsequent pulses, so that both holograms in a pair have consistent brightness. This suppression must also be tuned for different repetition rates.
By keeping the laser at 80 kHz and adjusting only the number of pulses and the camera/shutter timing, different inter-frame times can be achieved in steps of 12.5 µs without reopening the beam path or retuning the first-pulse suppression.

We have added this paragraph to section 2.2.2 to clarify: "For different timing schemes, it is practical to adjust the number of laser pulses in the burst and keep the frequency constant at 80 kHz, since first-pulse suppression settings and the neutral density filters in the beam path would otherwise need to be changed to compensate for the varying pulse energies at different repetition rates. Therefore, the timing can be freely adjusted by 12.5 µs steps."

*(C21): Line 194: It's unclear what simpleRTK2B and U-Blox ZED-F9P refers to.*

(A21): We thank the reviewer for pointing out the missing explanation and have added the information that it is a GNSS system: "For redundancy a multi-band, centimeter-level GNSS receiver board (the simpleRTK2B by ArduSimple, integrating the u-blox ZED-F9P modules), is also installed on HoloTrack, along with three GPS antennas. It is however currently not operational due to USB-interface issues in the current version."

*(C22): Line 244: "36 cm to the sample volume and are not look directly into…" is unclear, needs to be reworded.*

(A22): We have clarified this sentence and changed it to "Even with these upper bound assumptions, laser safety is guaranteed if operators do not come closer than 36 cm to the sample volume and do not look directly into the laser beam or direct reflections".

*(C23): Line 251: Provide a date and location for the IMPACT campaign.*

(A23): We thank the reviewer for this comment. The IMPACT campaign is now fully introduced (abbreviation, date and location at the beginning of Section 3:

"Firstly, during the IMPACT campaign ("In-situ Measurement of Particles, Atmosphere, Cloud and Turbulence" May-June 2025, Pallas Finland),…"

*(C24): Line 298: "degree" symbol is not properly coded in latex.*

(A24): We thank the reviewer and have fixed this mistake.

*C(25): Line 326: "Caused likely with 8-point-averaging filtering for 1D pitot tube" is unclear and grammatically incorrect… need to improve.*

(A25): We thank the reviewer for pointing this out and have changed the sentence to fix the mistake: "This is likely caused by the default 8-point-averaging filtering that was still set for 1D pitot tube recorder, effectively averaging out turbulence signal."

*(C26): Lines 330-331: Units for energy dissipation rate are incorrect.*

(A26): We thank the reviewer for catching this mistake and have changed the units to the correct $m^2/s^3$.

*(C27): Lines 335-338: The Stokes number is only one parameter that governs decoupling from the flow. The settling parameter also needs to be considered. I expect that for the larger drop diameter of 50 um the gravitational setting is an important contributor. Please discuss and evaluate.*

(A27): We thank the reviewer for raising this important point, we have added the settling parameter and a discussion about gravitational settling to the paragraph. Additionally we have corrected the Stokes numbers and changed the text in line 357 of the revised manuscript to:
"Even the larger dissipation found here of approximately $\varepsilon = 0.004$ $m^2/s^2$ would result in Stokes numbers of 0.005 and 0.13 (St = $\tau p/\tau K$ , where $\tau p = \rho_d d2/18\mu air$ is the particle response time and $\tau K = sqrt(pv_{air}/\varepsilon)$ is the Kolgomorov timescale) and settling parameters of 0.2 and 5 (Sv = ws\uk where ws = $(\rho d - \rho_{air}$ $gd^2)/18\mu air$ is the settling velocity and uK = $(v_{air}\varepsilon)^{1/4}$ the Kolmogorov velocity scale) for 10 μm and 50 μm diameter droplets respectively. In these conditions, we would expect even large cloud droplets to not have significant inertial effects (low St) but gravitational effects would be measurable. This deviation of droplets from the mean flow due to gravitational settling could be captured by HoloTrack. In more turbulent conditions or for even larger droplets, the strength of HoloTrack would be to observe the decoupling of larger droplets from the flow due to inertial effects as well."

*(C28): Line 350: "Recall" is not yet defined, so the sentence is not clear.*

(A28): We have added an explanation in that sentence "…recall, the rate of correctly detected and identified particles, is increased…." and a formal definition can be found in equation 5 in the following sub-section where recall is determined.

*(C29): Lines 361-364: I don't understand this, e.g., what is meant by "accurate measurement of precision". Accuracy and precision have two different meanings.*

(A29): We thank the reviewer for pointing out this potential source of confusion. By "precision," we mean the standard statistical definition as indicated by the definition in parentheses in the original manuscript. It is defined as the ratio of True Positives to Predicted Positives, which measures classification quality. We have added a clearer definition in the text and revised the sentence to: "…for accurate measurement

of precision (also known as positive predictive value, which indicates the fraction of detected droplets that are actually droplets and not False Positives)...".

*(C30): Figure 7: I don't recall that d_{gt} has been defined. What does the subscript indicate?*

(A30): We thank the reviewer for this comment. We have added "as a function of ground truth diameter d_{gt} of the printed circles" in the text in line 393 in the revised manuscript where Figure 7 is discussed and added in the caption of Figure 7 (Figure 8 in revised manuscript) "Recall measured with the CloudTarget as a function of the theoretical ground truth particle diameter d_{gt}." and "The results are binned by ground truth diameter d_{gt}, the diameter of the printed disks on the photomask which might vary slightly from the measured diameter d_m." for clarification.

*(C31): Lines 371-372: Is this a reasonable assumption? Why is it not possible to perform the analysis on an actual hologram pair?*

(A31): We thank the reviewer for the question. We consider this a reasonable assumption because position errors are expected to depend primarily on the z-position and constant from hologram to hologram. We have added "For this, we assume that the inter-particle distance between two particles in a single hologram is equivalent to the distance of one particle measured across two holograms of a hologram pair. Since we do not expect particle position error to vary significantly from hologram to hologram, this is a fair assumption." to line 400 of the revised manuscript.
In principle, one could design experimental setups to verify droplet displacement, and thus droplet velocity measurements, using a carefully calibrated laminar flow with small, neutrally buoyant droplets. However, the inherent inaccuracies of such an experiment are likely comparable to the small measurement uncertainties we aim to capture.
Therefore, only a static tool like CloudTarget is practical for this analysis. While it would be possible to displace CloudTarget using a motorized system with micrometer-level accuracy, measuring distances within a single hologram is equally effective and avoids additional uncertainties. The only uncertainty that can be mostly corrected is then the tilt of the target. The main factor that cannot be analyzed in the lab is any error introduced by movement of the camera arm relative to the sample volume, which is likely different during in-flight measurements. In-flight, this can be checked by observing whether static features, such as the background from the laser window or optical system, appear to move.

*(C32): Lines 373-374: s_m is used for both quantities... is this correct?*

(A32): This was not correct and a mistake by us, we thank the reviewer for pointing this out and have changed the one s_m to s_gt.

*(C33): Line 401: "weighted by the square root"... square root of what? And why?*

(A33): We thank the reviewer for the question. The weighting uses the square root of the droplet radius to include droplet size in the correlation while avoiding giving very small particles disproportionately low weight. In the images, the circle radii are scaled as $\sqrt{\text{actual\_radius}} \times 15 \times 10^{-6}$. We have made this clearer by making the subtle change from the to their in line 438 of the revised manuscript: "The particle sizes in these projections are artificially enlarged, weighted by their square root, to enhance weight of small particles in the correlation."

*(C34): Figure 9: The "small dark blue square" is difficult to find in the image. Please enhance and/or enlarge to make it more evident.*

(A34): Thank you for pointing this out. We cannot enlarge the square further, as it is intended to illustrate the correct scale. However, we have enhanced its visibility by changing the color to pink, using a wider line, and slightly increasing the overall figure size.

*(C35): Line 419: Do you mean "more turbulent data"?*

(A35): No, we mean data from experiments with a passive grid, where turbulence is weaker and it is easier to track the droplets. To phrase it more clearly, we have changed the sentence to "Here, we use the data from experiments with lower turbulence where we used an open i.e. passive grid as we expect our simple matching algorithm to be even more reliable in less turbulent flows."

*(C36): Line 426: Define PMR here so it is clearer what it indicates in Equation 7.*

(A36): PMR is already defined in Equation 6, just before Equation 7.

*(C37): Lines 432-440: I am completely lost. This discussion is quite tedious. Can it be simplified?*

(A37): We have rephrased parts of the section to clarify further, as this is however not a standard procedure the derivation is needed. To make the intent clearer and motivate the approach we have added the general idea of the approach before the derivation of equations 7 and 8 in line 460 of the revised manuscript: "The idea follows the simple argument, that only real particles are matched from hologram to hologram as noise that produces FP would not be displaced with approximately the mean velocity which is a requirement for matching. If the recall is known a certain PMR can be expected. If the PMR is lower than this expected value, we assume False Positives to be the cause."

*(C38): Line 447: I don't see that "both cases" are defined.*

(A38): We thank the reviewer for pointing this out and have rephrased the sentence more clearly to "The mean measured particle velocities and mean velocity measured by the 3D pitot tube agree remarkably well within an offset of less than 3.5% throughout the whole z range for two different mean velocities."

*(C39): Line 449: Is it really constant? It seems to me that it shows a trend.*

(A39): We have clarified the wording to: "Moreover, the measured velocity is **reasonably** constant throughout the whole z-range between the holographic arms ($z = 2.5$–$22$ cm), with slightly higher measured velocities for small z as can be seen in Figure 12." The small deviation at low z is close to the measurement uncertainty. If it were a systematic trend e.g. by influence of the arms, we would expect a symmetric pattern across the z-range, which is not observed. At this point we cannot identify a physical explanation for the minor offset.

*(C40): Line 458: Define turbulence intensity.*

(A40): To make a comparison between turbulence measurements and estimated uncertainties easier, we have replaced the use of turbulence intensity with the root-mean-square (RMS) of velocity fluctuations. The RMS of velocity fluctuations is now clearly defined in line 504 of the revised manuscript.

*(C41): Line 471: "Turbulence that exceeds the random error in inter-particle distances" is illogical. Please correct.*

(A41): We thank the reviewer for highlighting this. The sentence has been clarified to: "Mean velocity and velocity fluctuations that exceed the uncertainty in velocity measurement of 0.07m/s, can be accurately

measured in u (longitudinal) and w (vertical) direction." In the revision we now state the absolute uncertainty, as explained in the reply to the next comment.

*(C42): Figure 12: Are there results related to u versus w, which could hint at a sedimentation effect? It would be nice to include, in order to have a little bit of science in the article.*

(A42): We have added v and w component in Figure 13 and to the discussion in section 3.3.2. This demonstrates the higher uncertainty of the v component that was previously just mentioned. It also confirms the ability to measure w, as now discussed in the section. As also mentioned in line 527 of the revised manuscript, we do see a negative mean of w and a size dependency of the w-component: "The observed negative mean vertical velocity, stronger in the less turbulent case, hints toward sedimentation of the droplets in the wind tunnel experiments. This is further attested by the diameter dependent increasing negative w. Due to the limited data and to keep the scope of the paper clear, we refrain from further analyzing the relation in more detail here. Nontheless, it showcases HoloTrack's useful feature of measuring droplet dynamics and size at the same time to investigate the dependence." This comment, along with the investigation of *u* and *w*, has led us to reevaluate how we present velocity uncertainty in Section 3.2.2. Previously, we described it as a relative uncertainty as a function of the mean, which could be confusing. We have now clarified this by presenting it as a constant absolute error (which it also was before, therefore decreasing relative error. We believe this change makes the actual uncertainty more transparent.

*(C43): Line 484: "by the boundary layer"*

(A43): We have corrected this and added the article "the".

*(C44): Line 485: "expelled from the arms wake accumulate" is not clear… do you mean "from the wake of the arms"?*

(A44): We thank the reviewer for catching this error and have changed the sentence to "…the particles expelled from the wake of the arm accumulate…".

*(C45): Line 488: "regions are reach up to" should be "regions reach up to"?*

(A45): We thank the reviewer for noting this typographical error. It has been corrected to "regions reach up to".

*(C46): Lines 519-530: As noted above, this is all a repeat of what is in the article… not a Discussion. Similar in multiple other places in this section.*

(A46): We thank the reviewer for pointing this out and have removed these specific points from the discussion and restructured the whole discussion section as outlined in our response to comment C1.

*(C47): Lines 620-622: What does it mean that "HoloTrack would have sampled"? Is there in-cloud data or not? If so, please show it.*

(A47): We appreciate the reviewer raising this important point. We have now made clearer at several points in the manuscript, that the holographic system did record images during the maiden test flight (i.e. laser, camera, computer, automation all worked as expected) but due to an issue with the pinhole (now replaced) they were overexposed and hence no droplet data could be extracted from them. We clarified this in the following lines of the revised manuscript:
Line 265: "… HoloTrack had its maiden flight, successfully collecting various datasets, including holograms. Although a broken pinhole in the holographic optical system rendered the collected

holograms **too bright to be usable**, the test flight still demonstrated HoloTrack's ability to operate effectively under flight conditions."

Line 281: "As explained above, the pinhole used for spatial filtering of the laser beam was broken during the flight and in the campaign only the single short test flight was possible for HoloTrack**. Hence, for this test flight we could not evaluate in-situ holograms."**

Line 593: "As an example, **if the holograms in HoloTrack's maiden flight were not overexposed and could have been analyzed,** this would have…"

Line 629: As an example, based on yaw measurements and defining reliable unobstructed sub-volumes, we estimate ~34 L of high-precision holographic data would have been usable of the holograms in HoloTrack's maiden flight (if the holograms were not overexposed due to a broken pinhole)…."

Once again, we sincerely thank the reviewer for their very helpful and constructive comments. In response, we have made additional minor revisions to streamline the manuscript and enhance clarity. Furthermore, the CloudTarget analysis for $z$ = 12.2 has been improved with a more accurate tilt and "ghost-layer" (see Thiede 24, EGUsphere, 2025, 1–39, 2025b) correction. The velocity results in Figure 13 (previous Figure 12) are now shown for the final sub-volume, for consistency, which is also indicated in the revised manuscript. All other revisions are clearly indicated in the marked-up version of the manuscript.

---

## Author Response (AR2)

We would like to thank the editor for accepting the manuscript for publication. We also thank the two Reviewers for reviewing and positively assesing the revised manuscript. In the following we address the minor comments from Reviewer 2 on the revised manuscrip point by point. Additionally, we have changed the panel labels in figures 5, 7 and 13 to the correct style.

It is appreciated that the authors carefully and thoroughly addressed all prior comments. The following very minor corrections could be considered to improve the manuscript further, but in general I find the paper suitable for publication.

Figure 1: The addition of the new overview schematic and accompanying text is helpful in explaining the instrument concept. However, it's not clear to me whether it is intended that a shift in particle positions should be evident. The highlighted reference particle, and its change in position from the first to the second frame, is clear. Perhaps reduce the number of droplets if a shift in the pattern should be evident? (Now that I look at the figure closely again, I see that the blue highlighted droplets are shifted... but this is not explained in the caption, and again, I think perhaps the pattern could be made more evident.)

We have reduced the number of droplets to make the shifted droplet spatial pattern more obvious and added an explanation about the color scheme: "The droplets that are captured in both holograms of the pair, and for which the velocity can be calculated accordingly, are shown in blue. For the remaining droplets of each hologram, shown in green here, only size and position is known and velocity can not be determined."

Line 270: "Secondly, wo vital performance indicators" seems to have a typo or some missing information.

We thank the reviewer for catching this typo and have corrected this.

Line 376: "As detailed in Thiede et al. (2025b) recall, the rate of correctly detected and identified particles, is increased..." This sentence is somewhat cumbersome as written. Perhaps remove "recall", and delete comma between "particles" and "is increased".

We have changed it to "As detailed in Thiede et al. (2025b) the rate of correctly detected and identified particles (recall) is increased..."

Line 529: "This is further attested by the diameter dependent increasing negative w." The sentence is somewhat unclear. Does it mean that w becomes more negative as diameter increases?

We have changed the sentence to "This is further attested by the settling velocity (-w) increasing with droplet diameter.".

Line 617: The word "superhologram" is not defined, as far as I am aware. Either explain or rephrase so that the meaning of the sentence is clear.

Super-hologram is introduced, where it is first mentioned in Section 3.3.3. "As a first indication of influence of the holographic arms, specifically the tips, we show a "super-hologram" i.e. a heatmap of

relative concentration of detected droplets" but our inconsistent spelling made it hard to find, so we corrected it in line 617.